## PERSPECTIVE

# Redox-informed models of global biogeochemical cycles

Emily J. Zakem[1✉], Martin F. Polz[2,3] & Michael J. Follows[4]

Microbial activity mediates the fluxes of greenhouse gases. However, in the global models of the marine and terrestrial biospheres used for climate change projections, typically only photosynthetic microbial activity is resolved mechanistically. To move forward, we argue that global biogeochemical models need a theoretically grounded framework with which to constrain parameterizations of diverse microbial metabolisms. Here, we explain how the key redox chemistry underlying metabolisms provides a path towards this goal. Using this first-principles approach, the presence or absence of metabolic functional types emerges dynamically from ecological interactions, expanding model applicability to unobserved environments.

"Nothing is less real than realism. It is only by selection, by elimination, by emphasis, that we get at the real meaning of things." –Georgia O'Keefe

Microorganisms drive biogeochemical cycling in the earth system[1] (Fig. 1). Photo-autotrophic microorganisms are responsible for about half of $CO_2$ fixation and $O_2$ production on earth, and heterotrophic microorganisms are responsible for much of the return reaction: the oxidation of organic matter back into $CO_2$. The temporal and spatial separation of photoautotrophy and heterotrophy in the global environment drives the biological sequestration of carbon, the reduction of atmospheric $CO_2$, and the maintenance of elevated atmospheric and oceanic $O_2$[2–5]. Chemoautotrophic microorganisms also fix $CO_2$ and, together with anaerobic heterotrophic metabolisms, carry out diverse chemical transformations including the fluxes of nitrogen to and from biologically available states and the formation of the potent greenhouse gas nitrous oxide ($N_2O$)[6,7]. Since these transformations respond to, and feedback on, changes in climate (Fig. 1), estimating microbial activity accurately at global scales is important for climate science.

However, understanding and projecting the impacts of microbial processes are limited in part due to oversimplified representation in earth system models. For example, in marine biogeochemical models, much attention is given to the complex impacts of phytoplankton—the photoautotrophic microorganisms responsible for primary production—and their small zooplankton predators[8–10]. The bacterial and archaeal activities responsible for other critical aspects of biogeochemical cycling in the land and ocean—remineralization, denitrification, nitrogen fixation, methanogenesis, etc.—are often crudely parameterized[10]. Such models have limited prognostic capability. For example, models typically prescribe the ecological niche of a given

[1] Department of Biological Sciences, University of Southern California, Los Angeles, CA 90089, USA. [2] Department of Civil and Environmental Engineering, Massachusetts Institute of Technology, Cambridge, MA 02139, USA. [3] Department of Microbial Ecology, Center for Microbiology and Environmental Systems Science, University of Vienna, Vienna, Austria. [4] Department of Earth, Atmospheric, and Planetary Sciences, Massachusetts Institute of Technology, Cambridge, MA 02139, USA. ✉email: zakem@usc.edu

**Fig. 1 Key microbially driven redox transformations that mediate the atmospheric fluxes of climatically relevant gases.** Radiatively active gases are notated with red type. The processes in black type are represented in some way (though not necessarily with electron balancing) in both the marine and terrestrial biospheres in earth system models within the Coupled Model Intercomparison Project (land: NCAR Community Earth System Model[103]; ocean: GFDL COBALTv2[104]), which are used for projections of climate change in reports by the Intergovernmental Panel on Climate Change. Processes in green type are represented in only the terrestrial model. Current models do not yet include other relevant reactions, some of which are represented in gray type, such as anaerobic ammonia oxidation (anammox), the marine production and consumption of methane, the redox cycling of iron, manganese, and other metals, and the methane-relevant redox chemistry of phosphorus[105]. COBALTv2 does account for sulfate reduction in marine sediments, but sulfate is not represented. Image courtesy of NASA.

metabolism with imposed, empirically determined parameters that are site- or organism-specific. These parameterizations may or may not apply to other environments, including past and future ecosystems.

These simplistic approaches have been largely necessary due to the difficulties of characterizing the taxonomy and metabolic capabilities of natural microbial communities. However, the rapid expansion of genetic sequencing capabilities has enabled a clearer view of microbial biogeography and activity in the environment. In consequence, computational biogeochemistry is opening up the black box of remineralization and other microbially mediated processes in marine and terrestrial environments[11–18].

As we expand models to include the full metabolic potential of microorganisms, how can we organize and reduce the complexity of the descriptions of metabolic diversity? Non-photosynthetic organisms oxidize chemical species for energy, and thus their respiration is biogeochemically significant[19]. Here, we explain how the key reduction-oxidation (redox) reactions that supply energy for metabolisms can provide an additional organizing principle for explicit descriptions of microbial populations in ecosystem models. This redox basis can be exploited to quantitatively resolve chemical transformations in terms of assimilatory and respiratory fluxes. While not yet incorporated into earth system models, this view has been advocated for such applications[20], and has been embraced and employed in the field of

environmental biotechnology, such as in the interpretation and modeling of wastewater bioreactors[21]. Just as models of ocean and atmospheric circulation are constrained by conservation of energy and potential vorticity, complementing mass balance with powerful redox and energetic constraints enables self-consistent descriptions of diverse microbial metabolisms.

This approach aims to advance ecological modeling beyond species-specific descriptions to those that matter for biogeochemical function, in line with trait-based modeling approaches[9]. In analogy to the use of redox chemistry, trait-based functional type models of phytoplankton have used cell size as an organizing principle for understanding phytoplankton biogeography, biodiversity, and impact on biogeochemistry[9,22,23]. These types of theoretical constraints allow for the inclusion of more functional types without introducing as many degrees of freedom as would be necessary if each were empirically described. The guiding perspective is that organizing complex biological behavior by its underlying chemical and physical constraints gives more universally applicable descriptions of large-scale biogeochemistry.

When incorporating a redox-balanced approach into ecosystem models, microbial function emerges from underlying chemistry as a consequence of interactions between populations modeled as metabolic functional types and their environment. Resulting theoretically grounded ecosystem models independently simulate microbial growth, respiration, and abundances in

ways that we can compare with observations such as sequencing datasets. Thus, sequencing datasets are used as critical tests for the models, as external constraints rather than as input to the models, allowing for an iterative relationship between theory, observations, and models.

In contrast with empirically informed models, this approach involves constructing a model of microbial activity theoretically, and then comparing the results with the observations in order to gain an understanding of the system. The goal is to understand why biology functions as it does, in addition to anticipating global impacts. From a first-principles biogeochemical perspective with respect to physical and chemical forcing, genes are an intermediate step between forcing and function, with the detailed complexity of biological reality following the underlying chemical and physical constraints (analogous to the form follows function principle of architect Louis Sullivan). This does not equate to thinking that biology (or genetic information) does not matter or can be replaced entirely in models by physics and chemistry. Rather, we want to fundamentally understand biological activity as an integrated part of an ecosystem, and physics and chemistry become tools for doing so.

Here, we outline the basis for using redox chemistry as an organizing principle and its translation into quantitative descriptions of microbial activity that are simple enough for global earth system models. We then discuss the benefits of this approach in the context of their implications for improved understanding and projections of global change impacts. Finally, we discuss limitations and possible future developments.

**Predicting microbial activity.** From one perspective, microbial communities are characterized by interactions at the micro-scale: gene expression, enzymatic capabilities, metabolites, species-specific interdependencies, etc., as well as the physical and chemical environment surrounding small cells[24–28]. The information from sequencing in particular has allowed for a huge expansion of insight into the detailed in situ activity of uncultivated species. When investigating global-scale impacts, how do we decide which of these details may be bypassed for simplicity? Or, if this simplification is impossible, must we incrementally construct a microbial ecosystem model that incorporates all known micro-scale detail?

Another way forward arises from a macro-scale perspective, which examines how ecosystem function relates to the chemical potential utilized by organisms for energy[20,29,30]. For example, it is well known that microbial communities in sediments and anoxic zones organize according to the redox tower – the ranking of half-reactions by electrochemical potential[14,31,32]. Furthermore, respiration by living organisms increases the entropy of the environment by dissipating concentrated sources of chemical energy in accordance with the Second Law of Thermodynamics[30,33,34].

This perspective suggests that chemical potential can be used to predict the activity of microbial communities and their biogeochemical impact. However, given the notorious complexity of microbial cells and systems, which is many steps away from governing chemical or physical equations, how can we be sure that this activity is indeed predictable? Frentz et al.[35] demonstrated that external conditions cause the seemingly random fluctuations observed in microbial growth, rather than stochastic variation in gene expression. This provides direct evidence of deterministic behavior, and so the authors conclude that microbial systems can in principle be determined by macroscopic laws.

How is this determinism manifested? If microbial communities can respond relatively quickly to changes to their local environment, they may predictably optimize the exploitation of locally available resources. In the ocean, dispersal in microbes is thought to be a highly efficient process such that microbial communities can in effect draw from an extensive seed bank[36,37], as captured in the phrase "everything is everywhere, the environment selects"[38]. Furthermore, recent evidence also shows that gene acquisitions and deletions happen quickly enough to allow for horizontal gene transfer to dominate bacterial adaptation[39–43], implying that evolution can occur within few generations and thus on timescales similar to ecological interactions. Perhaps consequentially, similar geochemical environments have been demonstrated to have high-microbial functional redundancy despite different taxonomic compositions[17,44]. This may be interpreted with the hypothesis that physics and chemistry selects for metabolic traits, and that these traits can be housed in different organisms with taxonomic composition shaped by micro-scale or biotic interactions[17,44,45].

The prediction of microbial activity from environmental chemical potential has a long history in microbiology[20,21,46–52], and is conceptually similar to other redox-balanced approaches to understanding microbial activity in sediments, soils, subsurfaces, and aquatic systems[13,31,53–58]. Illustrating the power of these approaches, anticipating metabolism from chemical potential resulted in a prediction that anaerobic ammonia oxidation (anammox) should exist decades before it was observed[59,60]. Quantitatively understanding microbially mediated rates of conversion of substrates has practical implications for wastewater treatment, and thus the field of biotechnology has established methodologies for an approach in textbook form[21]. Flux balance analysis (FBA) models can be considered as much more highly detailed analogs of this approach that resolve the mass and electron balances among a multitude of chemical reactions within a cell[61,62].

**Redox-balanced metabolic functional types.** We can resolve microbial activity in global ecosystem models using the underlying redox chemistry of diverse metabolisms as a constraint. One specific way forward is to model distinct metabolisms as populations of metabolic functional types. This systematically quantifies relative rates of substrate consumption, biomass synthesis, and excretions of transformed products associated with each metabolism. Coupled with estimates of substrate uptake, this replaces implicit parameterizations of processes such as organic matter consumption, oxygen depletion, and denitrification with electron-balanced respiratory fluxes of dynamic microbial populations. Box 1 provides a detailed description of this methodology for multi-dimensional models.

A particular set of redox reactions may distinguish a functional type, such as the oxidation of organic matter using oxygen (aerobic heterotrophy), or the oxidation of ammonia or nitrite using oxygen (chemoautotrophic nitrification) as exemplified in Table 1. For each metabolism, an electron-balanced description consists of multiple half-reactions: biomass synthesis, oxidation of an electron donor, and reduction of an electron acceptor[21,48]. The ratio of anabolism and catabolism can then be represented by the fraction $f$ of electrons fueling cell synthesis vs. respiration for energy, following ref. [21]. This provides a yield $y$ (moles biomass synthesized per mole substrate utilized) of each required substrate that reflects two inputs: electron fraction $f$ and the coefficients of the half-reactions (Fig. 2). The interlinked yields reflect the energy supplied by the redox reaction, the energy required for synthesis and other cellular demands, and the inefficiencies of energy conversion. Either $f$ or $y$ for any one of the substrates may be estimated theoretically with Gibbs free energies of reaction[21] or with a combination of theoretical and empirical strategies[63].

---

**Box 1  Incorporating metabolic functional types into ecosystem models**

A metabolic functional type can be represented as a population with a growth rate that is limited or co-limited by multiple required substrates. If Liebig's Law of the Minimum is employed, the limiting growth rate $\mu$ is described as

$$\mu = \min(V_i y_i) \tag{1}$$

where $V_i$ is the specific uptake rate of substrate $i$, and yield $y_i$ is the biomass yield with respect to that substrate. Yields for the different substrates and elements are interlinked in the metabolic budget derived from the underlying redox chemistry. Yields reflect Gibbs free energies of reaction among other factors. In the simplest model, non-limiting substrates are consumed in proportion to the limiting resource according to the metabolic budget, although in reality they may accumulate in the form of storage molecules.

Each metabolic functional type population can be incorporated into a multi-dimensional environmental model (e.g., an ocean simulation) with physical transport as

$$\frac{dB}{dt} = \mu B - L(B)B - \underbrace{\nabla \cdot (\mathbf{u}B)}_{\text{advection}} + \underbrace{\nabla \cdot (\boldsymbol{\kappa} \nabla B)}_{\text{diffusion}} \tag{2}$$

for biomass concentration $B$, loss rate $L$, velocity $\mathbf{u}$, and diffusion coefficient $\boldsymbol{\kappa}$. The loss rate function varies with biomass and represents a combination of processes, including predation, viral lysis, maintenance, and senescence. These processes remain largely unconstrained, although efforts have been made to relate losses to ecological dynamics[107,108].

The yield partitions the amount of substrate taken up by the population into that used for growth, $V_i y_i$, versus that exiting the cell in modified form as a waste product, $V_i(1 - y_i)$ (Fig. 2 and Table 1). Equation 1 suggests a correlation between $\mu$ and $y$, but yields may be further modified by other factors. For example, accounting for maintenance energy decreases the ratio of growth to respiration, contributing to a decoupling between growth rate and yield particularly at low growth rates[109]. Furthermore, a trade-off between uptake rate and yield at the cellular level reflects the allocation of enzyme towards machinery for substrate uptake vs. biomass synthesis, among other factors. Considering a proteome constraint can incorporate this trade-off (Supplementary Note 1).

---

**Table 1 Simplified equations describing two exemplary metabolic functional types.**

|  | Aerobic heterotroph | Ammonia-oxidizing chemoautotroph |
|---|---|---|
| $R_D$ (1) | $\frac{1}{d_D} C_{c_D} H_{h_D} O_{o_D} N_{n_D} \rightarrow \frac{n_D}{d_D} NH_4^+ + \frac{c_D}{d_D} CO_2 + H^+ + e^-$ | $\frac{1}{6} NH_4^+ + \frac{1}{3} H_2O \rightarrow \frac{1}{6} NO_2^- + \frac{4}{3} H^+ + e^-$ |
| $R_E$ (1-$f$) | $\frac{1}{4} O_2 + H^+ + e^- \rightarrow \frac{1}{2} H_2O$ | $\frac{1}{4} O_2 + H^+ + e^- \rightarrow \frac{1}{2} H_2O$ |
| $R_S$ ($f$) | $\frac{n_B}{d_B} NH_4^+ + \frac{c_B}{d_B} CO_2 + H^+ + e^- \rightarrow \frac{1}{d_B} C_{c_B} H_{h_B} O_{o_B} N_{n_B}$ | $\frac{n_B}{d_B} NH_4^+ + \frac{c_B}{d_B} CO_2 + H^+ + e^- \rightarrow \frac{1}{d_B} C_{c_B} H_{h_B} O_{o_B} N_{n_B}$ |
| $R_T$ | $\frac{1}{d_B} C_{c_B} H_{h_B} O_{o_B} N_{n_B} + \frac{(1-f)}{4} O_2 \rightarrow \frac{f}{d_B} B + \left(\frac{n_D}{d_D} - \frac{n_B f}{d_B}\right) NH_4^+ + \left(\frac{c_D}{d_D} - \frac{c_B f}{d_B}\right) CO_2$ | $\left(\frac{1}{6} - \frac{f}{d_B}\right) NH_4^+ + \frac{c_B f}{d_B} CO_2 + \frac{(1-f)}{4} O_2 \rightarrow \frac{f}{d_B} B + \frac{1}{6} NO_2^-$ |
| $e^-$ donor yield | $y_D = f \frac{d_D}{d_B} \approx f$ | $y_{NH_4^+} = \left(1 + \frac{d_B}{6f}\right)^{-1} \approx \frac{6f}{d_B}$ |
| Example efficiency | $f = 0.1 - 0.2$; Marine bacteria[101] | $f = 0.02 - 0.04$; Marine archaea[18] |
| Example budget | $7.1 C_{6.6} H_{10.9} O_{2.6} N + 47 O_2 \rightarrow B + 6.1 NH_4^+ + 42 CO_2$ | $112 NH_4^+ + 5 CO_2 + 162 O_2 \rightarrow B + 111 NO_2^-$ |

For each, half-reactions combine to form the catabolic and anabolic full reactions[21]: the oxidation of an electron donor ($R_D$; here either organic matter or ammonium), the reduction of an electron acceptor ($R_E$), and biomass synthesis ($R_S$). The total reaction ($R_T$) sums each of these three multiplied by a factor of $f$, the fraction of electrons partitioned into the synthesis reaction vs. respiration. Denominator $d$ represents the number of electron equivalents that correspond to the oxidation states of the inorganic constituents of that synthesis: with a microbial biomass composition of $C_5H_7O_2N$, $d_B = 4(5) + 1(7) - 2(2) - 3(1) = 20$. Organic matter oxidation and synthesis equations are written without $H_2O$ on the left- and right-hand side, respectively, for conciseness. Charge balance via speciation of DIC is also neglected for simplicity. Example whole organism metabolic budgets are calculated using the listed example efficiencies for marine organisms, an average marine organic substrate composition[102] of $C_{6.6}H_{10.9}O_{2.6}N$, and the above biomass composition.

---

The result is a stoichiometric budget of the metabolism of the whole organism (Table 1). These descriptions quantify the elemental ratios of utilized substrates, biomass, and the excretion of waste products. For example, the descriptions account for the $CO_2$ produced by heterotrophic metabolisms as well as the $CO_2$ fixed by chemoautotrophic metabolisms (Table 1, Fig. 2, and Supplementary Fig. 2), linking microbial activity directly to global carbon cycling.

To estimate the growth rate of each functional type, the yields from the metabolic budgets are combined with the uptake rates of the required substrates (Box 1). Limiting uptake rates may rely on empirically derived uptake kinetic parameters, or they can be estimated theoretically from diffusive supply, cell size, membrane physiology, and other physical constraints[64–66]. If theoretical models of uptake are used, the physical constraints on substrate acquisition and the redox chemical constraints on energy acquisition can provide an entirely theoretical estimate of the growth of each metabolic functional type.

One strategy is to represent the populations carrying out each of these discrete metabolisms as one functional type population, which aggregates the diverse community of many species that are fueled by the same (or a similar) redox reaction (Fig. 3). Such aggregation has been deemed a useful strategy for representing the biogeochemical impacts of microbial communities for certain research questions[67,68]. However, for other questions this wipes out critical diversity among the aggregated populations. For example, diverse aerobic heterotrophic populations consume organic matter over a wide range of rates, and these rates dictate the amount of biologically sequestered carbon in the ocean. Redox chemistry and physical limitations alone may not inform the heterogeneity among similar metabolisms. One additional constraint is the limited capacity of the cell and thus its allocation of proteome towards different functions[69]. While the electrons supplied to the cell must be conserved following the redox balance, the electrons may be partitioned differently into machinery for substrate uptake vs. biomass synthesis, for instance, for different phenotypes. This partitioning can be quantitatively related to ecological fitness and biogeochemical impact via uptake kinetics, effective yields, and other traits[9] (Supplementary Note 1 and Supplementary Fig. 1).

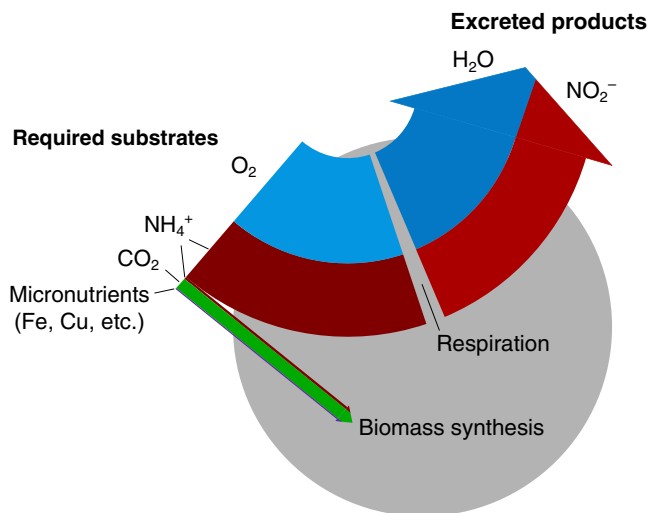

**Fig. 2 Schematic of a single cell represented as a metabolic functional type carrying out the aerobic oxidation of ammonia.** The redox balance informs the elemental ratios of substrates utilized, biomass synthesized, and waste products excreted (Table 1).

**Benefits and implications for anticipating global change.** Redox chemistry aids in reducing the number of degrees of freedom in descriptions of diverse microbial metabolisms. We next discuss the benefits of this electron-balanced approach, each contextualized by specific projected impacts of global change due to microbial activity and broad challenges in the fields of microbial ecology and biogeochemistry.

*Flexible and broadly applicable metabolic thresholds*: A key question for microbial biogeochemical studies, for which biogeochemical models are primed to answer, is how the biogeographies of diverse, active metabolisms vary with changes in the physical and chemical environment. What threshold determines the viability of a given metabolism?

Redox-balanced metabolic budgets obviate the need to impose critical concentrations or other thresholds that determine the presence of any given metabolism. Rather than being imposed following empirical relationships, metabolic biogeography emerges dynamically from ecological interactions and reflects environmental chemical potential. This flexibility aids in understanding metabolic thresholds more fundamentally, and it expands model applicability to diverse and unobserved environments. This is of particular importance for understanding global change, since past and future worlds may include very different ecosystems that do not reflect current empirical trends.

For example, the oceans are currently losing oxygen due to global warming[5,70]. If anoxic zones continue to expand, this will increase the habitat of anaerobic microorganisms, whose respiration results in emissions of $N_2$ and $N_2O$ to the atmosphere[7]. Many biogeochemical models prescribe $O_2$ concentrations that inhibit anaerobic activity in accordance with observations of specific organisms or communities in experimental conditions. This assumes that the same $O_2$ concentrations limit metabolism similarly in all environments, and often trades mechanistic understanding of oxygen limitation for empirical correlations that may reflect a variety of natural and introduced biases, such as micro-scale heterogeneity, physical mixing in the ocean, and experimental bottle effects.

In contrast, a metabolic functional type model does not require imposed oxygen threshold concentrations (Supplementary Fig. 3). When oxygen supply is abundant, anaerobic types are competitively excluded because growth using alternative electron acceptors is lower than with oxygen. When oxygen supply is

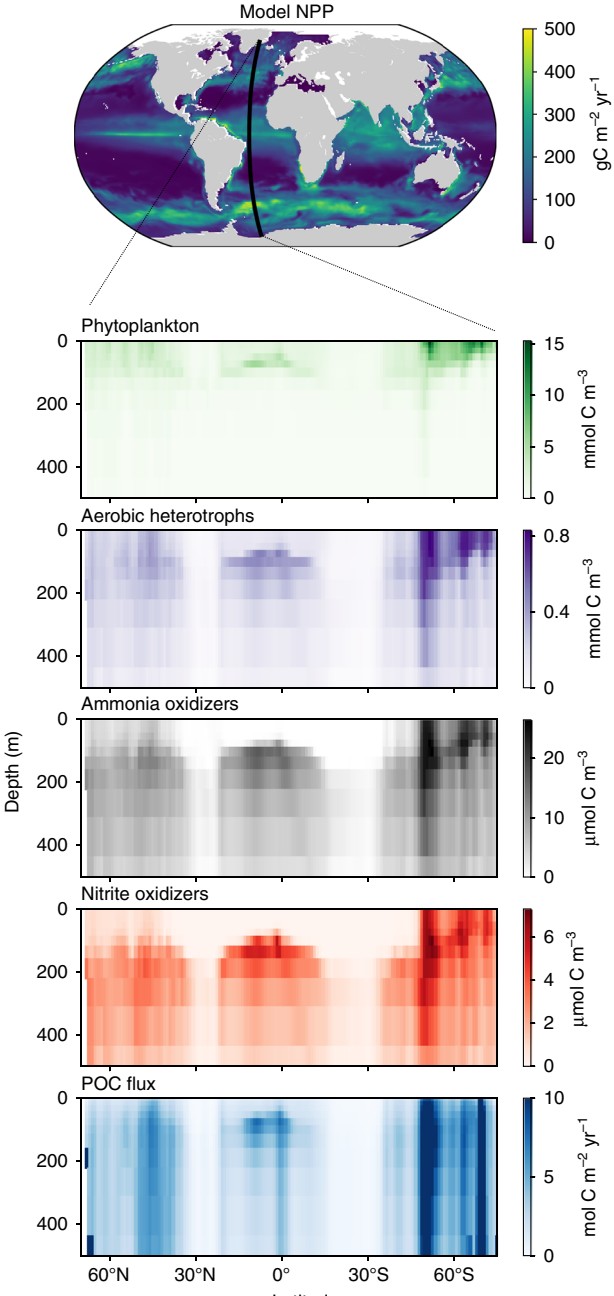

**Fig. 3 Solutions from a global simulation resolving multiple metabolic functional types.** Net primary productivity (NPP), the biomasses of the metabolic functional types, and the sinking particulate organic carbon (POC) flux are resolved along a transect of a global microbial ecosystem model coupled with an estimate of the ocean circulation (Darwin-MITgcm[18]).

low, aerobic populations may persist and continue to deplete any available oxygen even as their growth is limited by oxygen, allowing for a steady state stable coexistence of aerobic and anaerobic metabolisms, which is consistent with a variety of observations[71].

Descriptions of microbial growth that reflect underlying chemical potential can enable predictions of many other metabolic transitions, such as nitrogen fixation, nitrification, and the transition to sulfur oxidation and reduction[13,18,72–74]. As another example, this approach predicts the restriction of

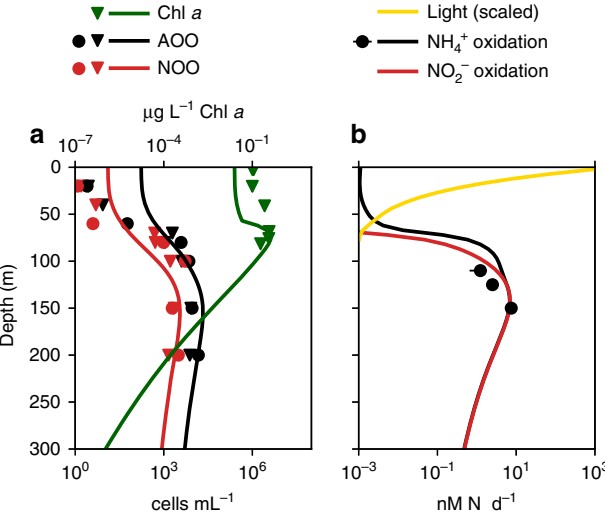

**Fig. 4 Model simulation and observations of the marine nitrification system.** Biogeochemistry is driven by microbial metabolic functional types in a vertical water column model[18]. Lines are model solutions, and marked points are observations from two stations in the Pacific Ocean[75,106] (see Supplementary Fig. 2 for more detail) (**a**). Chlorophyll *a* concentrations and abundances of ammonia-oxidizing organisms (AOO) and nitrite-oxidizing organisms (NOO). Observed abundances are of the 16S rRNA abundances of archaeal Marine Group I and *Nitrospina*-like bacteria[75,106]. Model abundances are converted from biomass with 0.1 fmol N cell[-1] for AOO, 0.2 fmol N cell[-1] for NOO[76], and one gene copy per cell. (**b**). Light (solar irradiance) and bulk nitrification rates.

nitrification from the sunlit surface ocean as a consequence of competitive exclusion by phytoplankton in many environments (Fig. 4 and Supplementary Fig. 2), as well as active nitrification in some surface locations where phytoplankton are limited by another factor not affecting the chemoautotrophs, such as at high latitudes where phytoplankton are limited by light[18]. The emergent exclusion from most of the surface ocean anticipates that many clades of nitrifying microorganisms have adapted to long-term exclusion from the surface and consequentially lost (or did not develop) photoprotective cellular machinery.

*Replacing implicit descriptions of organic matter remineralization*: The fate of organic matter dictates the amount of carbon sequestered in the marine and terrestrial biospheres. Microbial consumption mediates the carbon stored in soils, the carbon stored in the ocean as dissolved organic matter (DOM), and the sinking flux of organic carbon that constitutes the marine biological carbon pump[4], without which atmospheric $CO_2$ would be 100–200 ppm higher than current levels. We want to understand how these carbon reservoirs respond to changes in climate, such as increased temperatures and changes in precipitation patterns. However, in biogeochemical models, simple rate constants often dictate the remineralization of elements from organic back into inorganic constituents.

Replacing simplistic parameterizations with dynamic metabolic functional types means that electron-balanced descriptions of growth and respiration instead drive the fate of organic matter in earth system models (Fig. 3). In addition to a more sophisticated and responsive description of carbon sequestration, non-living organic matter is fully integrated into ecosystem frameworks, enabling theoretical studies of phytoplankton-bacteria interactions to complement observational and experimental approaches.

Much work remains in the development of these descriptions. As we discuss below, accurate estimates of organic matter turnover rates require more accurate descriptions of the complex

processes governing microbial uptake rates of organic matter. However, the redox-informed yields are still useful for quantifying the relative amount of $CO_2$ excreted and the absolute amount of biomass sustained on a given substrate, independent of uptake kinetics (Supplementary Note 2).

*Relationships between abundances, rates, nutrient concentrations, and elemental ratios*: An overarching puzzle challenging microbial ecology is to understand how chemical transformations in the environment are set by the ecological interactions at the organism level, among individual microscopic cells. It is clear that abundances of populations are not simply and directly correlated with biogeochemical impact (i.e., higher abundance does not necessarily imply an associated higher rate of chemical transformation). Untangling the relationship between abundances and biogeochemical function is also necessary for interpretation of genetic evidence that provide insight into this complex ecosystem structure.

Redox-balanced metabolic functional type modeling links rates of biomass synthesis associated with a particular metabolism to its rate of respiration as well as the standing stock of limiting nutrients. As functional type modeling is coupled with estimates of population loss rates due to grazing, viral lysis, or other mortality, simulations also resolve the standing stocks of functional biomass. This quantifies the relationship between biomass concentrations and volumetric rates of chemical transformations, emphasizing how relatively low biomass may be associated with relatively high bulk rates[71].

For example, the approach has revealed a clear example of the signature of chemical potential in the ecology of marine nitrification[18] (Fig. 4). In this model, the two steps of nitrification are represented by two functional type populations. This predicts about a three-fold difference in the abundances of the organisms responsible for each of the two steps of nitrification, despite the fact that the two populations carry out the same rate of subsurface N-cycling at steady state[6,18]. A three-fold or greater difference in abundance and associated ammonium ($NH_4^+$) and nitrite ($NO_2^-$) concentrations is consistent with observed differences[18,75], and it reflects that the oxidation of one mole of $NH_4^+$ generates three times more electrons than the oxidation of one mole of $NO_2^-$, with differences in cell size further contributing to differences in abundances (Fig. 4 and Supplementary Fig. 2). Recent observations confirm the redox-based difference in $NH_4^+$ and $NO_2^-$ biomass yield[76,77], although measured rates from a nonsteady environment suggest that $NO_2^-$-oxidizing bacteria can partition electrons more efficiently than $NH_4^+$-oxidizing archaea[76] (i.e., higher fraction *f* despite lower yield *y*; see Supplementary Note 3).

As redox-based descriptions resolve the stoichiometry of whole organism metabolism, they also link together elemental cycles. Explicit description of relative elemental flow through the ecosystem, and specifically their variation from average values, is critical for understanding climate-biogeochemical feedbacks[78–80]. For example, the nitrification model also estimates the $CO_2$ fixation rates associated with nitrification rates (Supplementary Fig. 2), enabling global-scale, electron-balanced projections of the amount of carbon converted to organic form by chemoautotrophic nitrifying microorganisms.

*Connections with sequencing datasets*: How do we relate metabolic functional type models to sequencing datasets measuring genetic, transcriptomic, and proteomic diversity? Connecting biogeochemical models with sequencing data is critical because this data provides an enormous amount of information about ecosystem structure and function. Genes (or transcripts) themselves are not necessarily the most concise or useful currency given functional redundancies, unattributed function, and variation in gene dosage from horizontal gene

transfer as well as growth rate[40]. Recent gene-centric models aim to resolve the abundances of key genes as proxies for a predetermined set of metabolic pathways[13,14,17]. However, the parameters used to describe metabolic pathways in these models are estimated similarly to the redox-balanced yields and efficiencies described here.

The innovation of gene-centric models is the sophisticated conversion of estimates of biogeochemical activity and biomass to genes. For example, the model of Coles et al.[17] resolves biomass and nutrient concentrations prognostically, and then uses a three-part formula—representing constitutive, regulated, and steady state transcription—to diagnostically calculate transcription rates from modeled biomass and growth rates[17]. Thus, the two types of modeling are complimentary, with redox chemistry providing estimates of metabolic activity from fundamental principles, and the careful calibrations between activity and sequencing providing a comparative metric.

The examples here externalize the conversion between modeled activity and sequencing information as a transparent process. In Fig. 4, the predicted functional biomass of ammonia-oxidizing population is related to archaeal Marine Group I (MGI) and *Nitrospina*-like 16S rRNA genes with two conversion factors: the cell elemental quota (fmol $N$ cell$^{-1}$) and the number of cellular gene copies. Conversion error arises since cell mass and size vary with growth rate[81,82]. Maintaining transparency of the conversion from predicted microbial activity to genes and transcripts allows interdisciplinary audiences to understand and critique the models.

**Limitations and possible extensions**. Using chemical potential as a theoretically grounding organizing principle for the resolution of diverse metabolisms can greatly improve microbial descriptions in global biogeochemical models. However, the approach does have its limitations, which generally increase in significance with increased temporal or spatial resolution.

Modeling metabolic diversity with functional type populations requires choosing how metabolisms are distributed among the populations. This has consequences when interpreting time-varying states: model solutions become dependent on the partitioning of metabolism among the functional types as the timescales of physical change approach the timescales of microbial growth (see Supplementary Note 4, Supplementary Fig. 3, and Supplementary Fig. 4 for a detailed example). Other species-specific time-varying phenomena such as the lag response of organisms to substrate availability also become relevant[83]. On one hand, this is beneficial for resolution of microbial processes in fine-grained ocean circulation models where flow can vary on the order of days. However, incorporating another constraint, such as proteome allocation[69], is necessary to inform these choices. For example, considering enzymatic allocation in combination with energetics allowed for the prediction of both the division of nitrification into a two-step process in mixed environments and the combined, complete pathway in one organism (comammox) in biofilms, which preceded observations of the latter[84–86].

Uncertainty in distributions of metabolism lies not only in the length of a metabolic pathway, but also in the degree of metabolic versatility (metabolic mixotrophy). Such versatility characterizes key players in large-scale biogeochemistry, such as nitrite-oxidizing bacteria and photoheterotrophs[87–89]. Mixotrophic lifestyles can increase the fitness of populations in their environments, impacting overall ecosystem function[90]. In one sense, the approach here provides a prediction of where we might expect such mixotrophy by resolving stable coexistences of diverse metabolisms. In Fig. 3, for example, syntrophic coexistence occurs at depth among heterotrophs, ammonia oxidizers,

and nitrite oxidizers, and future work could investigate what determines which combinations of these coexistences remain as passive interactions, which develop into mutualistic dependencies as active interactions[91], and which evolve into mixotrophic phenotypes or endosymbionts. Additionally, by considering the potential to carry out a metabolism as a trait, we can use the current framework along with an additional constraint to investigate implications of metabolic mixotrophy. For example, Coles et al.[17] impose a trade-off between the degree of metabolic diversity of a single functional type and growth rate, enabling the exploration the consequences of distribution of metabolism on the biogeochemical state.

Also, the metabolic functional type approach resolves only active functional biomass, while evidence suggests that less than 10% to more than 75% of the microbial community may be inactive[92]. Some seemingly inactive populations may slowly metabolize over long timescales, requiring longer model integration times and careful attention to their loss rates for resolution, while some populations are periodically active as revealed by high-resolution observations in time[37].

The proposed modeling approach relies on estimates of the limiting uptake rates of required substrates. In lieu of suitable theoretical descriptions, the use of empirically derived uptake kinetic parameters still employs the benefits of the redox-informed yields (Supplementary Note 2). However, underlying physical constraints to substrate acquisition can in principle be exploited to develop more universally applicable descriptions for a variety of substrates and contexts. Uptake kinetics are complex, but for many limiting resources, encounter effectively controls the uptake, and the physics of encounter has been relatively well described. For example, uptake rates estimated from diffusive supply of substrate, cellular geometry, and membrane physiology[64–66] have been empirically supported[93]. For organic matter, future work is needed to develop suitable descriptions of consumption rates, whether empirical or theoretical. For example, descriptions require attention to the hydrolysis of organic compounds by extracellular enzymes and the ecology of sinking marine particles—the diffusion of monomer away from the particle, within-particle transport, and dynamic ecological interactions on particle surfaces, among other processes[11,74,94,95].

In Fig. 3, the electron-balanced description consists of an average stoichiometry and electron fraction for one sinking pool of organic matter in the ocean, which, as mentioned above, is not sufficient to accurately resolve the carbon storage that is shaped by a distribution of turnover rates. As one of the many factors impacting the rates, an energetics-based perspective can serve as a tool for further deciphering organic matter complexity. For example, organic matter may be partially organized by the nominal oxidation state of its carbon atoms, which relates to a measure of free energy and accessibility[96,97]. This could be used to improve the phenomenological description of organic matter in models as labile vs. non-labile, for example, with a more mechanistic underpinning.

Descriptions of phytoplankton are currently much more sophisticated than of bacteria and archaea in models, reflecting a longer history of comprehensive sets of observations. However, further work could develop simple descriptions of photoautotrophic metabolisms from underlying energetics by connecting the supply of photons to available energy for biosynthesis within the cell. Many biogeochemical models account for an inefficiency of phytoplankton metabolism with a parameter that dictates their excretions of dissolved organic matter[98]. Incorporating this excretion into an energetic framework would enhance studies of phytoplankton ecology, such as studies of photoautotrophic-heterotrophic interactions in the ocean surface or photoautotrophic-chemoautotrophic interactions at the base of

the euphotic zone where some phytoplankton excrete nitrite due to incomplete reduction of nitrate[99].

As a more radical extension, can we progress past population modeling and model microbial consortia as one aggregate community biomass[34,100]? This may improve resolution of time-varying metabolic versatility. However, if both steps of nitrification were a part of such a consortium, would the characteristic accumulation of nitrite be predicted (Supplementary Fig. 2)? We leave these questions for future research and conclude that the best choice for the degree of resolution of metabolism will depend on the specific research question and the available observations.

We have described a useful approach for understanding and anticipating microbial control of biogeochemical cycling that is suitable for global applications. The approach aims to represent microbial growth and respiration explicitly and consistently from knowledge of chemical gradients in the environment, towards a goal of building an independently constructed theoretical ecosystem model that can then be compared to observations. Describing microbial communities with underlying energetic constraints connects metabolisms dynamically with global geochemical distributions, such as those of carbon dioxide, oxygen, and biologically available nitrogen. This deepens our understanding of microbial ecosystems and enables the incorporation of the feedbacks of microbial activity to changes in global biogeochemistry and the climate system.

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

## Acknowledgements

We thank Alyson Santoro for providing data and Deepa Rao for providing the introductory quote. We are grateful for helpful feedback on Fig. 1 from Andreas Richter, Victoria Orphan, and Naomi Levine. We appreciate discussions over the past years with Amala Mahadevan, Dan Repeta, Penny Chisholm, Joe Vallino, and Terry Hwa that have influenced this perspective. E.J.Z. was supported by the Simons Foundation (Postdoctoral Fellowship in Marine Microbial Ecology). M.F.P. was supported by a grant from the Simons Foundation (LIFE ID 572792), M.J.F. was supported by the Simons Foundation: the Simons Collaboration on Ocean Processes and Ecology (SCOPE #329108) and the Simons Collaboration on Computational Biogeochemical Modeling of Marine Ecosystems (CBIOMES #549931).

## Author contributions

E.J.Z. wrote and revised the manuscript. M.F.P. and M.J.F. contributed to revising and editing the manuscript.

## Competing interests

The authors declare no competing interests.
