## [Peer Review File · Nature Communications]

Reviewers' comments:

Reviewer #1 (Remarks to the Author):

Zakem et al is an interesting perspective piece with an up and coming early career researcher in the lead authorship position. The manuscript advocates the use of redox chemistry based theory for the development of mechanistic ocean models that include microbial biogeochemistry. While not novel (see for example their references 14,15,17), this concept is rarely invoked in ocean modeling and in earth system model communities who have not adopted some of the soil and microbial engineering concepts. I agree with the potential in the approach, and that it is deserving of attention and novel to its intended audience.

I find the discussion of how using redox chemistry to remove arbitrary thresholds and create emergent metabolic biogeography extremely compelling. (L204 section)

In general, the figures are not as compelling as the text. I have made some specific recommendations below, but for example, Figure 4 is an extremely complex set of panels, each of which has 4-6 model lines as well as several observational points. The purpose of the figure is to illustrate that nitrification is excluded from the euphotic zone due to competition and thus does not need to be parameterized as inhibited in light environments. But the figure can't tell this story with this level of complexity. It would be great to see just the key elements in one figure or cartoon. Note that figure 4 is invoked later to also show an important point that the energy available from oxidizing ammonia vs nitrite differs, and thus can explain biomass differences in the organisms performing this transformation. But I think this could still be included in a single figure or conceptual diagram. Similarly, Figure 3 is visually more appealing, but it doesn't make a particularly compelling case for the argument. E.g. the figure doesn't contrast the difference in POC flux for a redox based vs. traditional model.

I disagree with the authors in section 4 on gene fluent predictions. They state that resolving genes does not constrain models. In fact, I think this contradicts their prior section on the role of biomass vs rate of turnover of chemical constituents. Genes and transcripts while poor proxies for biomass and rates do constrain the model as demonstrated in their Figure 4b. Additionally, while genes and transcripts are a poor substitute for rate measurements, they do reflect an instantaneous sample of the community function unbiased by bottle or incubation effects, and far more rapid and feasible than counting cells. Thus, they make an excellent constraint for models, and are becoming ever more widely available as data sets for comparing with models. This in no way suggests that redox chemistry as suggested here is not an excellent strategy for future modeling efforts. I don't object to many of their points in this section. The suggestion that the ocean is pure physics and redox chemistry is certainly excitingly provocative. But as the authors are well aware, biological mediation of this effort (via genetic codes for proteins etc) will fail in regions where biologically mediated processes are subject to secondary constraints. Perhaps the most obvious of these being iron limitation, but one could also include nitrogen fixation in nitrogen replete coastal zones which is energetically inefficient.

As mentioned in the specific comments, there are a few concepts which dilute the impact of the key message. Although they are interesting, and have been central in the research groups of the authors, they distract from the central theme (e.g. L65, Box 2).

Specific comments:

- L65: I think this concept is a bit of a distraction, though I agree with the content. Its sets up two paragraphs beginning with a question in a row, and the second question is the main concept of this perspective, so an aside about trait based modeling as size without further elaboration just distracts.
- Abstract: L26 -29 is a little bit too terse for comprehension – particularly by a general audience. I'd recommend to delete sentence "Ongoing" L20 and use the space to express what these benefits mean.

- L19: replace “predictions” with “projections” see nice discussion here: <https://sciencepolicy.colorado.edu/zine/archives/1-29/26/guest.html>
- Figure 1: Perhaps sulfate reduction is included in Cobalt, but this raises some questions about what “represented” is. Sulfur is not included as a substrate, so the sulfate reduction is the residual after all the organic matter has been oxidized by other processes. I would consider this a grey area (literally). On the other hand, N₂ is not tracked, but diazotrophy is “represented” so perhaps the authors are correct.
- Figure 1: I am not expert in the CESM land model but it clearly has some representation of photosynthesis. https://escomp.github.io/ctsm-docs/versions/release-clm5.0/html/tech_note/Photosynthetic_Capacity/CLM50_Tech_Note_Photosynthetic_Capacity.html#rst-photosynthetic-capacity. Perhaps the authors could check with someone about whether they would agree with this characterization. There are clearly stored carbon pools in the land model, and this must be a result of photoautotrophy.
- Figure 1: I think your argument is actually strengthened by noting that so many of these processes are really not represented. Perhaps the radiatively active gasses that are in the atmosphere could be called out in a color... I.e. CO₂, CH₄, H₂O, N₂O? This would make the link back to climate forcing stronger.
- L94: In reviewing Kiorboe et al, 2018, I note that their “resource acquisition” does include metabolic pathways for acquiring energy, and they call out both photosynthesis and organic matter oxidation. Still, I like this turn of phrase, perhaps the authors might explore how to note that resource acquisition is in fact metabolic potential writ larger than just heterotrophy and photoautotrophy.
- L98: Figure 1 is not very related to this paragraph. I might perhaps fit better in Paragraph 1 of the overview and then be further referenced. To me the figure most demonstrates the inter-related fluxes of radiatively active gasses between ocean, land, and atmosphere mediated by microbes.
- Box 1: Is this too much information? What is perhaps new is the concept of yield and estimation of the link between yield and uptake rate. You haven’t chosen to discuss here the literature on connecting Gibbs energy from the reaction to estimation of the yield. (as in reference 14, or for example Smeaton and Van Cappellen 2018 *Geochimica et Cosmochimica Acta* and references within)
- Table 1: equation for RT ammonia oxidizers has a typo I believe. It should be $(1/6 + f/db) \text{NH}_4$, reflecting the need for additional N to synthesize biomass, and this then gives the listed yield.
- L178: This might be where it would be useful to at least point to approaches for estimating energetic inefficiency.
- Box 2: Here, I feel the discussion goes a bit off track. You have argued for using chemical equilibria and mechanistic underpinnings to understand microbial communities. But now you prescribe traits that are linked to fundamental cellular processes as an arbitrary theoretical tradeoff. I think it makes more sense to describe what physiological factors might lead to this tradeoff in terms of allocating cell resources, genetic streamlining or other mechanistic processes. After all, energy (or at least electrons) must be conserved as the redox chemistry indicates. Your f “efficiency” in Table 1 is tied to this concept too I think, and perhaps that could be woven into the discussion.
- Reference 110 needs a journal
- L308: This sentence got a little bit convoluted. Perhaps a more direct statement?
- L 372. This is a great point for linking organic matter complexity to free energy. I would find it interesting to see this expanded on. Perhaps this could be an important strategy for characterizing organic matter in some more physically meaningful way than high molecular and low molecular weight for example as proxies for “labile” and “non labile”.
- L383: I think this is untrue, most BGC models do not assume a yield of 1 for nitrogen substrates for example. E.g. Fasham, Ducklow, McKelvie, *J Mar. Res.*, Vol. 48, pp. 591–639, 1990. CESM1, Dynamic Green model.
- L402, stylistically, I think I prefer communities. As community, implies one single entity. Perhaps this is the point?
- Figure 2, I think maybe this figure could be more illustrative. Respiration should probably occur within the cell, rather than outside it as implied by the word placement. Would it be useful to consider the differences between energy electron donor substrates and other mass substrates? So organic matter might be black, and O₂ and/or Fe or other micro/macronutrients might be red. Instead of y ,

maybe yield could be used to simplify the mental conversion effort. Of course there may be no other nutrients required, so I see the issue.

Victoria Coles

Reviewer #2 (Remarks to the Author):

This manuscript is well-written and I mostly agree with the authors' suggestions. However, the suggestions that the authors present are not particularly novel. Mostly, they are advocating that models explicitly represent the functional groups responsible for multiple biogeochemical transformations, many of which are often implicitly modeled in current biogeochemical models (e.g., an implicit function that relates nitrification rate to ammonium concentrations, without explicitly modeling nitrifying bacteria). I think that most biogeochemists would agree that explicit models of more diverse metabolic processes are important to accurately simulate the response of mesopelagic and seafloor microbial communities to climate change. However, the difficulties have typically been derived from 1) a dearth of information necessary to constrain these processes and 2) the need to explicitly include (and parameterize) many different functional groups while simultaneously including much greater vertical resolution in the deep ocean.

The suggestion for the inclusion of more functional groups expressing wider metabolic capabilities is certainly not novel to this manuscript. Indeed multiple studies have added additional functional groups to the more common autotroph based communities in epipelagic food webs. Studies by Bouskill et al. (2012, *Front. Microbiol.*), Bianchi et al. (2018, *Nat. Geo.*), Coles et al. (2017, *Science*), and Reed et al. (2014, *PNAS*) all come to mind. The authors seem mostly to be arguing that these sorts of studies should be re-branded as "Redox-informed" models, rather than simply as plankton functional group models that include additional metabolic processes.

The novelty in this study comes from their suggestion of using the yield of different metabolic reactions as a structuring principle for determining microbial growth rates and hence the distribution of metabolic processes across a model ocean. While this seems like a powerful idea initially, there are some distinct difficulties in utilizing this approach in practice. As the authors note in Equation 2 of Box 1, the growth rate of an organism will be determined by the uptake rate of that organism for that substrate multiplied by the yield of that substrate. However, the uptake rate is not known a priori or determinable based on theory. In most simple modeling frameworks, the growth rate is modeled as a saturating function of the limiting substrate concentration multiplied by a maximum growth rate. From what I can tell of the authors' approach, they would model the growth rate as a maximum uptake rate multiplied by the yield multiplied by a saturating function. Mathematically, this seems equivalent to the typical approach, although the standard approach subsumes the yield into the maximum growth rate (which is unimportant if the maximum uptake rate cannot be determined a priori).

While I think it is important to explicitly incorporate more metabolic strategies into biogeochemical models as the authors recommend, I am not sure that this discussion has really enlightened me. I would, however, be very eager to see the results if they successfully develop the type of model that they are describing.

Specific comments:

Lines 70-73: The authors are relating their approach to the approach of size-structured models. However, more recent advances in size-structured modeling highlight some of the difficulties associated with such approaches. For instance, in the Ward et al. (2012) model the ideal of a single allometric group of phytoplankton gave way to four different taxonomic groups (each modeled allometrically), the three smallest of which only really existed in the model ocean over a linear size range of approximately a factor of 2. This was a useful advance that was necessary because larger

taxa (e.g., diatoms) grow at faster rates than similarly sized cyanobacteria. However, with this necessary added complexity the number of free parameters is no longer substantially reduced relative to models that simply include four types of phytoplankton. I suspect that similar issues would be encountered with the “redox-informed” modeling approach as each functional group would have to have its uptake rates carefully tuned.

Lines 233 – 237: This is not a particularly strenuous test of whether or not a redox-balanced approach is useful. The importance of competition with phototrophs for ammonium in limiting nitrification in surface oceans has been shown by, for instance Smith et al. (2014, PLOS One) and Xu et al. (2019, L&O) and is very easily to simulate mechanistically as long as ammonium is produced through remineralization of organic matter in both the euphotic and disphotic zones and phototrophs are given higher uptake rates for ammonium than nitrifiers.

Lines 249 – 252: I completely agree that the use of a prescribed “Martin Curve” is an inaccurate way to quantify carbon flux through the mesopelagic. My understanding is that it is seldom used except when computational limitations preclude explicit modeling of the mesopelagic ocean (e.g., in mixed layer or euphotic zone only models).

Lines 327-328: The question of whether genes are “intermediaries” or actively modeled is largely dependent on whether the investigator has a biogeochemical or ecological standpoint. From the biogeochemical standpoint, the gene is, in fact, only an intermediary that reflects the metabolic potential of the organism. However, from an ecological standpoint competition and selection are acting at the level of the organism (or even at the level of the gene given the prevalence of horizontal gene transfer). When considered from this perspective the gene is not an intermediary, but instead the gene (and the organism that contains it) are quite active.

Draft

Response to Reviewer Comments

Reviewer #1 (Remarks to the Author):

Zakem et al is an interesting perspective piece with an up and coming early career researcher in the lead authorship position. The manuscript advocates the use of redox chemistry based theory for the development of mechanistic ocean models that include microbial biogeochemistry. While not novel (see for example their references 14,15,17), this concept is rarely invoked in ocean modeling and in earth system model communities who have not adopted some of the soil and microbial engineering concepts. I agree with the potential in the approach, and that it is deserving of attention and novel to its intended audience.

I find the discussion of how using redox chemistry to remove arbitrary thresholds and create emergent metabolic biogeography extremely compelling. (L204 section)

Thank you! We took this into account when choosing what to highlight in the new, shortened abstract.

In general, the figures are not as compelling as the text. I have made some specific recommendations below, but for example, Figure 4 is an extremely complex set of panels, each of which has 4-6 model lines as well as several observational points. The purpose of the figure is to illustrate that nitrification is excluded from the euphotic zone due to competition and thus does not need to be parameterized as inhibited in light environments. But the figure can't tell this story with this level of complexity. It would be great to see just the key elements in one figure or cartoon. Note that figure 4 is invoked later to also show an important point that the energy available from oxidizing ammonia vs nitrite differs, and thus can explain biomass differences in the organisms performing this transformation. But I think this could still be included in a single figure or conceptual diagram.

Thanks for this suggestion. The old Fig. 4 was indeed very complicated. We have revised Fig. 4 into a much simpler set of plots, with the more detailed version in the Supplement (Fig. S1). We still needed two panels to show the two points.

Similarly, Figure 3 is visually more appealing, but it doesn't make a particularly compelling case for the argument. E.g. the figure doesn't contrast the difference in POC flux for a redox based vs. traditional model.

We agree that it does not make the point that the POC flux is an improvement, but we use it to support a different point in the main text (L. 262-264): simply that it is feasible to recapitulate all of these fluxes as electron-balanced respiratory fluxes of explicitly modeled populations. We haven't done the work yet to actually demonstrate any improvement (see below points on the heterogeneity of populations and organic matter constituents of the POC flux). We have removed the sentence "The POC flux is determined entirely by the modeled populations of

microorganisms and its sinking velocity” to avoid an implication that the flux simulation has improved quantitatively relative to observations.

I disagree with the authors in section 4 on gene flux predictions. They state that resolving genes does not constrain models. In fact, I think this contradicts their prior section on the role of biomass vs rate of turnover of chemical constituents. Genes and transcripts while poor proxies for biomass and rates do constrain the model as demonstrated in their Figure 4b. Additionally, while genes and transcripts are a poor substitute for rate measurements, they do reflect an instantaneous sample of the community function unbiased by bottle or incubation effects, and far more rapid and feasible than counting cells. Thus, they make an excellent constraint for models, and are becoming ever more widely available as data sets for comparing with models.

We realize we were not clear in our discussion of constraints. We meant that the genes do not provide input into the specific type of modeling approach that we are describing. We agree wholeheartedly that they can and should be used with this approach as external constraints to compare against model simulations. We have revised the text throughout the manuscript to articulate our position:

L 77-79: “These types of theoretical constraints allow for including more functional types without introducing as many degrees of freedom as would be necessary if each were empirically described.”

L 88-90: “Thus, sequencing datasets are used as critical tests for the models, as external constraints rather than as input to the models, allowing for an iterative relationship between theory, observations, and models.”

This in no way suggests that redox chemistry as suggested here is not an excellent strategy for future modeling efforts. I don't object to many of their points in this section. The suggestion that the ocean is pure physics and redox chemistry is certainly excitingly provocative. But as the authors are well aware, biological mediation of this effort (via genetic codes for proteins etc) will fail in regions where biologically mediated processes are subject to secondary constraints. Perhaps the most obvious of these being iron limitation, but one could also include nitrogen fixation in nitrogen replete coastal zones which is energetically inefficient.

We have made some changes to this section to better communicate that we think sequencing information is of critical use, and that we do not think that biology can be ignored entirely. We've added the following paragraph to the overview:

L 91-101: “In contrast with empirically informed models, this approach involves constructing a model of microbial activity theoretically, and then comparing the results with the observations in order to gain an understanding of the system. The goal is to understand why biology functions as it does, in addition to anticipating global impacts. From a first-principles biogeochemical perspective with respect to physical and chemical forcing, genes are an intermediate step between forcing and function, with the detailed complexity of biological reality following the underlying chemical and physical constraints (analogous to the “form follows function” principle of architect Louis Sullivan). This does not equate to thinking that biology (or genetic

information) does not matter or can be replaced entirely by physics and chemistry. Rather, we want to fundamentally understand biological activity as an integrated part of an ecosystem, and physics and chemistry become tools for doing so.”

While we have left the paragraph about how electron-balancing can complement “gene-centric” approaches (beginning on L. 315), we no longer refer specifically to “gene-fluent” approaches. Benefit 4 is now headed as “**Connections with sequencing datasets**” (L. 305). We have removed most of the paragraph that included the term “deprioritizing genes,” to avoid misinterpretation of the intent.

As mentioned in the specific comments, there are a few concepts which dilute the impact of the key message. Although they are interesting, and have been central in the research groups of the authors, they distract from the central theme (e.g. L65, Box 2).

We agree, and we have now removed these concepts for sharper focus. Specifically, we have removed the previous L65 paragraph and Box 2 (as detailed below in response to the specific comments).

Specific comments:

- L65: I think this concept is a bit of a distraction, though I agree with the content. Its sets up two paragraphs beginning with a question in a row, and the second question is the main concept of this perspective, so an aside about trait based modeling as size without further elaboration just distracts.

Thank you for this suggestion. We’ve removed this paragraph to get to our point more quickly without distraction. We’ve retained one point from this paragraph – that cell size is analogous to redox chemistry in its utility for model development – in revised form in a later paragraph:

L 73-82: “This approach aims to advance ecological modeling beyond species-specific descriptions to those that matter for biogeochemical function, in line with trait-based modeling approaches¹⁰. In analogy to the use of redox chemistry, trait-based functional type models of phytoplankton have used cell size as an organizing principle – a “master trait” – for understanding phytoplankton biogeography, biodiversity, and impact on biogeochemistry^{10,26–30}. These types of theoretical constraints allow for including more functional types without introducing as many degrees of freedom as would be necessary if each were empirically described. The guiding perspective is that organizing complex biological behavior by its underlying chemical and physical constraints gives more universally applicable descriptions of large-scale biogeochemistry.”

- Abstract: L26 -29 is a little bit too terse for comprehension – particularly by a general audience. I’d recommend to delete sentence “Ongoing” L20 and use the space to express what these benefits mean.

Excellent point, thank you. We had to shorten the abstract to 100 words, and so we have taken this advice by cutting both the “Ongoing” sentence and most of previous lines L26-29. We

retained one benefit (the emergent metabolic biogeography), based on your above note, and explained it more thoroughly.

- L19: replace “predictions” with “projections” see nice discussion here: <https://sciencepolicy.colorado.edu/zine/archives/1-29/26/guest.html>

Thank you for this suggestion. We have replaced as recommended here and throughout the text when in used conjunction with future/climate modeling. We retained the term prediction when discussing the approach more abstractly. We changed to “simulations” when talking about our model output specifically. And thanks for the link to the discussion. (EJZ has taken a science policy class at CU Boulder so was especially excited to have this connection!)

- Figure 1: Perhaps sulfate reduction is included in Cobalt, but this raises some questions about what “represented” is. Sulfur is not included as a substrate, so the sulfate reduction is the residual after all the organic matter has been oxidized by other processes. I would consider this a grey area (literally). On the other hand, N₂ is not tracked, but diazotrophy is “represented” so perhaps the authors are correct.

Yes, it is a grey area! It makes sense to draw the line where relevant substrates are resolved, so we’ve changed to grey as recommended. (Also the sulfate reduction is for the sediments, so one could argue that it is not part of the marine biosphere.) Even though N₂ is not tracked, DIN is, and so we’ll keep diazotrophy in black. To avoid any confusion, we added the following to the caption:

L 769-770: “COBALTv2 does account for sulfate reduction in marine sediments, but sulfate is not represented.”

- Figure 1: I am not expert in the CESM land model but it clearly has some representation of photosynthesis. https://escomp.github.io/ctsm-docs/versions/release-clm5.0/html/tech_note/Photosynthetic_Capacity/CLM50_Tech_Note_Photosynthetic_Capacity.html#rst-photosynthetic-capacity. Perhaps the authors could check with someone about whether they would agree with this characterization. There are clearly stored carbon pools in the land model, and this must be a result of photoautotrophy.

Thank you, good point. This was an oversight on our end – we were focusing on microbial processes, but it is unnecessarily confusing to exclude terrestrial photosynthesis in this way. We’ve changed this to black.

- Figure 1: I think your argument is actually strengthened by noting that so many of these processes are really not represented. Perhaps the radiatively active gasses that are in the atmosphere could be called out in a color... Ie. CO₂, CH₄, H₂O, N₂O? This would make the link back to climate forcing stronger.

Great idea! Thank you very much. We’ve changed the radiatively active gases to red font and noted in the caption.

• L94: In reviewing Kiorboe et al, 2018, I note that their “resource acquisition” does include metabolic pathways for acquiring energy, and they call out both photosynthesis and organic matter oxidation. Still, I like this turn of phrase, perhaps the authors might explore how to note that resource acquisition is in fact metabolic potential writ larger than just heterotrophy and photoautotrophy.

Thank you for pointing this out. When taking into account the above advice to tone down discussion of phytoplankton, cell size, trait-based models, etc., we’ve decided to remove this sentence all together. Now, this paragraph simply points out the analogy between cell size and redox chemistry, and how they both can contribute to organizing parameters:

L 73-82: “This approach aims to advance ecological modeling beyond species-specific descriptions to those that matter for biogeochemical function, in line with trait-based modeling approaches¹⁰. In analogy to the use of redox chemistry, trait-based functional type models of phytoplankton have used cell size as an organizing principle – a “master trait” – for understanding phytoplankton biogeography, biodiversity, and impact on biogeochemistry^{10,26–30}. These types of theoretical constraints allow for including more functional types without introducing as many degrees of freedom as would be necessary if each were empirically described. The guiding perspective is that organizing complex biological behavior by its underlying chemical and physical constraints gives more universally applicable descriptions of large-scale biogeochemistry.”

• L98: Figure 1 is not very related to this paragraph. I might perhaps fit better in Paragraph 1 of the overview and then be further referenced. To me the figure most demonstrates the inter-related fluxes of radiatively active gasses between ocean, land, and atmosphere mediated by microbes.

We have moved the first reference to Fig. 1 to the first paragraph of the overview, and have removed it from this paragraph.

• Box 1: Is this too much information? What is perhaps new is the concept of yield and estimation of the link between yield and uptake rate. You haven’t chosen to discuss here the literature on connecting Gibbs energy from the reaction to estimation of the yield. (as in reference 14, or for example Smeaton and Van Cappellen 2018 *Geochimica et Cosmochimica Acta* and references within)

Yes, it also seemed too much upon rereading. We have revised and shortened Box 1, and have included the reference to Gibbs free energies and the above 2018 citation.

• Table 1: equation for RT ammonia oxidizers has a typo I believe. It should be $(1/6+f/db) NH_4$, reflecting the need for additional N to synthesize biomass, and this then gives the listed yield.

Thank you for catching this typo.

- L178: This might be where it would be useful to at least point to approaches for estimating energetic inefficiency.

We have revised this paragraph to better communicate f vs. y , and have included citations for estimating the efficiencies. The updated second half of the paragraph is:

L 170-178: The ratio of anabolism and catabolism can then be represented by the fraction f of electrons fueling cell synthesis vs. respiration for energy, following Rittman and McCarty 2001²⁵. This provides a yield y (moles biomass synthesized per mole substrate utilized) of each required substrate that reflects two inputs: electron fraction f and the coefficients of the half-reactions (Fig. 2). The interlinked yields reflect the energy supplied by the redox reaction, the energy required for synthesis and other cellular demands, and the inefficiencies of energy conversion. Either f or y for any one of the substrates may be estimated theoretically with Gibbs free energies of reactions²⁵ or with a combination of theoretical and empirical strategies⁷⁵.

- Box 2: Here, I feel the discussion goes a bit off track. You have argued for using chemical equilibria and mechanistic underpinnings to understand microbial communities. But now you prescribe traits that are linked to fundamental cellular processes as an arbitrary theoretical tradeoff. I think it makes more sense to describe what physiological factors might lead to this tradeoff in terms of allocating cell resources, genetic streamlining or other mechanistic processes. After all, energy (or at least electrons) must be conserved as the redox chemistry indicates. Your f “efficiency” in Table 1 is tied to this concept too I think, and perhaps that could be woven into the discussion.

We agree that Box 2 was off track. We have moved the main points to Supplementary Text 1. Also, we have woven in a discussion of how both electrons and physiological factors can be taken into account via the use of proteome allocation as a constraint. The revised paragraph is:

L 190-204: “One strategy is to represent the populations carrying out each of these discrete metabolisms as one functional type population, which aggregates the diverse community of many species that are fueled by the same (or a similar) redox reaction (Fig. 3). Such aggregation has been deemed a useful strategy for representing the biogeochemical impacts of microbial communities for certain research questions^{79,80}. However, for other questions this wipes out critical diversity among the aggregated populations. For example, diverse aerobic heterotrophic populations consume organic matter over a wide range of rates, and these rates dictate the amount of biologically sequestered carbon in the ocean. Redox chemistry and physical limitations alone may not inform the heterogeneity among similar metabolisms. One additional constraint is the limited capacity of the cell and thus its allocation of proteome towards different functions⁸¹. While the electrons supplied to the cell must be conserved following the redox balance, the electrons may be partitioned differently into machinery for substrate uptake vs. biomass synthesis, for instance, for different phenotypes. This partitioning can be quantitatively related to ecological fitness and biogeochemical impact via uptake kinetics, effective yields, and other traits¹⁰ (Supplemental Text S1 and Fig. S2).”

- Reference 110 needs a journal

Journal added.

- L308: This sentence got a little bit convoluted. Perhaps a more direct statement?

We have shortened to:

L 537: “The examples here externalize the conversion between modeled activity and sequencing information as a transparent process.”

- L 372. This is a great point for linking organic matter complexity to free energy. I would find it interesting to see this expanded on. Perhaps this could be an important strategy for characterizing organic matter in some more physically meaningful way than high molecular and low molecular weight for example as proxies for “labile” and “non labile”.

We are glad you found it interesting. We have expanded to include:

L 380-381: “This could be used to replace the phenomenological description of organic matter in models, as “labile” vs. “non-labile,” for example, with a more mechanistic underpinning.”

- L383: I think this is untrue, most BGC models do not assume a yield of 1 for nitrogen substrates for example. E.g. Fasham, Ducklow, McKelvie, J Mar. Res., Vol. 48, pp. 591--639, 1990. CESM1, Dynamic Green model.

Thank you for noting this. We have changed this paragraph to reflect that models do often incorporate a DOM excretion by phytoplankton (which we mistakenly thought was only for DOC), and to rephrase to suggest that this excretion be linked to the energetic framework:

L 387-396: “Descriptions of phytoplankton are currently much more sophisticated than of bacteria and archaea in models, reflecting a longer history of comprehensive sets of observations. However, further work could develop simple descriptions of photoautotrophic metabolisms from underlying energetics by connecting the supply of photons to available energy for biosynthesis within the cell. Many biogeochemical models account for an inefficiency of phytoplankton metabolism with a parameter that dictates their excretions of dissolved organic matter¹¹². Incorporating this excretion into an energetic framework would enhance studies of phytoplankton ecology, such as studies of photoautotrophic-heterotrophic interactions in the ocean surface or photoautotrophic-chemoautotrophic interactions at the base of the euphotic zone where some phytoplankton excrete nitrite due to incomplete reduction of nitrate^{113,114}.”

- I402, stylistically, I think I prefer communities. As community, implies one single entity. Perhaps this is the point?

We agree “communities” is better and have changed here, as well as in a few other places throughout the text.

- Figure 2, I think maybe this figure could be more illustrative. Respiration should probably occur within the cell, rather than outside it as implied by the word placement. Would it be useful to consider the differences between energy electron donor substrates and other mass substrates? So organic matter might be black, and O₂ and/or Fe or other micro/macronutrients might be red. Instead of y, maybe yield could be used to simplify the mental conversion effort. Of course there may be no other nutrients required, so I see the issue.

Thank you for these suggestions. We have reworked the schematic accordingly. It now describes the budget of an ammonia oxidizing functional type, rather than a generic type.

Victoria Coles

Draft Only

Reviewer #2 (Remarks to the Author):

This manuscript is well-written and I mostly agree with the authors' suggestions. However, the suggestions that the authors present are not particularly novel. Mostly, they are advocating that models explicitly represent the functional groups responsible for multiple biogeochemical transformations, many of which are often implicitly modeled in current biogeochemical models (e.g., an implicit function that relates nitrification rate to ammonium concentrations, without explicitly modeling nitrifying bacteria).

We are advocating a specific approach for the explicit representation: the use of redox chemistry to reduce the number of degrees of freedom in the model.

I think that most biogeochemists would agree that explicit models of more diverse metabolic processes are important to accurately simulate the response of mesopelagic and seafloor microbial communities to climate change. However, the difficulties have typically been derived from 1) a dearth of information necessary to constrain these processes and 2) the need to explicitly include (and parameterize) many different functional groups while simultaneously including much greater vertical resolution in the deep ocean.

The suggestion for the inclusion of more functional groups expressing wider metabolic capabilities is certainly not novel to this manuscript. Indeed multiple studies have added additional functional groups to the more common autotroph based communities in epipelagic food webs. Studies by Bouskill et al. (2012, *Front. Microbiol.*), Bianchi et al. (2018, *Nat. Geo.*), Coles et al. (2017, *Science*), and Reed et al. (2014, *PNAS*) all come to mind. The authors seem mostly to be arguing that these sorts of studies should be re-branded as "Redox-informed" models, rather than simply as plankton functional group models that include additional metabolic processes.

There are many ways in which metabolic functional types may be incorporated into ecosystem models. Here, we are suggesting one particular way, rooted in underlying chemistry, which can replace the reliance on ad hoc experimental or observational data. We have cited all of these studies in the manuscript.

The novelty in this study comes from their suggestion of using the yield of different metabolic reactions as a structuring principle for determining microbial growth rates and hence the distribution of metabolic processes across a model ocean.

This is indeed the main point. We have modified the organization of the introduction to get to this main point more quickly:

We have removed the previous paragraph on cell size, which allows us to start the discussion of redox chemistry in the fourth paragraph.

L 63-65: “Here, we explain how the key reduction-oxidation (redox) reactions that supply energy for metabolisms can provide an additional organizing principle *for explicit descriptions of microbial populations* in ecosystem models.” (The italicized text is new.)

L 77-82: “These types of theoretical constraints allow for including more functional types without introducing as many degrees of freedom as would be necessary if each were empirically described. The guiding perspective is that organizing complex biological behavior by its underlying chemical and physical constraints gives more universally applicable descriptions of large-scale biogeochemistry.”

While this seems like a powerful idea initially, there are some distinct difficulties in utilizing this approach in practice. As the authors note in Equation 2 of Box 1, the growth rate of an organism will be determined by the uptake rate of that organism for that substrate multiplied by the yield of that substrate. However, the uptake rate is not known a priori or determinable based on theory. In most simple modeling frameworks, the growth rate is modeled as a saturating function of the limiting substrate concentration multiplied by a maximum growth rate. From what I can tell of the authors’ approach, they would model the growth rate as a maximum uptake rate multiplied by the yield multiplied by a saturating function.

Mathematically, this seems equivalent to the typical approach, although the standard approach subsumes the yield into the maximum growth rate (which is unimportant if the maximum uptake rate cannot be determined a priori).

Growth rate is indeed treated as the convolution of uptake and yield so $\mu = Vy$, where V is the limiting uptake rate. However, we contend that deconstructing μ into mechanistic components is valuable, and not equivalent to simply estimating the growth rate directly. Both uptake and yield can be modeled to some extent from basic principles (physics of encounter is important for the former, and redox balance for the latter). Hence the separation allows us to bring to bear fundamental principles including conservation laws (energy and electrons) that are not necessarily respected by an imposed μ . It is true that uptake kinetics are somewhat complex, but for the limiting resource, when drawn down to limiting concentrations, encounter effectively controls the uptake, and the physics of encounter has been relatively well described. Transporter allocation at high uptake rates is less well constrained, but allometric scalings for uptake kinetics are based on clear geometric principles and are empirically supported (e.g. Litchman et al, 2007; Aknes and Egge, 1991 as cited within the manuscript).

These points were not stated clearly enough in the previous version. We have now included discussion of this point in two places in the text:

L 184-189: “To estimate the growth rate of each functional type, the yields from the metabolic budgets are combined with the uptake rates of the required substrates (Box 1). Limiting uptake rates can be estimated theoretically from diffusive supply, cell size, membrane physiology, and other physical constraints⁷⁶⁻⁷⁸. Together, physical constraints on substrate acquisition and redox chemical constraints on energy acquisition can provide an entirely theoretical estimate of the growth of each metabolic functional type.”

L 336-340: “The proposed modeling approach relies on estimates of the limiting uptake rates of required substrates. Uptake kinetics are complex, but for the limiting resource, encounter effectively controls the uptake, and the physics of encounter has been relatively well described. For example, uptake rates estimated from diffusive supply of substrate, cellular geometry, and membrane physiology⁷⁶⁻⁷⁸ have been empirically supported⁹⁸. ”

The significance of the yields is that they quantify the respiratory fluxes, and not solely the growth rate of the populations. Thus, the CO₂ excreted by a heterotroph or the NO₂⁻ excreted by an ammonia oxidizer, for example, can be linked to the electron budgeting of the metabolism. We have added a few sentences to clarify this:

L 180-183: “These descriptions quantify the elemental ratios of utilized substrates, biomass, and the excretion of waste products. For example, the descriptions account for the CO₂ produced by heterotrophic metabolisms as well as the CO₂ fixed by chemoautotrophic metabolisms (Table 1, Fig. 2, Fig. S1), linking microbial activity directly to global carbon cycling.”

While I think it is important to explicitly incorporate more metabolic strategies into biogeochemical models as the authors recommend, I am not sure that this discussion has really enlightened me. I would, however, be very eager to see the results if they successfully develop the type of model that they are describing.

Note that Figs. 3, 4, and S2 are examples of this type of model.

Specific comments:

Lines 70-73: The authors are relating their approach to the approach of size-structured models. However, more recent advances in size-structured modeling highlight some of the difficulties associated with such approaches. For instance, in the Ward et al. (2012) model the ideal of a single allometric group of phytoplankton gave way to four different taxonomic groups (each modeled allometrically), the three smallest of which only really existed in the model ocean over a linear size range of approximately a factor of 2. This was a useful advance that was necessary because larger taxa (e.g., diatoms) grow at faster rates than similarly sized cyanobacteria. However, with this necessary added complexity the number of free parameters is no longer substantially reduced relative to models that simply include four types of phytoplankton. I suspect that similar issues would be encountered with the “redox-informed” modeling approach as each functional group would have to have its uptake rates carefully tuned.

In a follow-up study using the model of Ward et al. (2012), Ward et al. (2014) showed that top-down control allowed for diversity in size within each of the four taxonomic groups. However, we have now removed the paragraph on size-structured models because, in light of comments from both reviewers, we agree that it distracts from the main point. The new paragraph now reads as:

L 73-82: “This approach aims to advance ecological modeling beyond species-specific descriptions to those that matter for biogeochemical function, in line with trait-based modeling approaches¹⁰. In analogy to the use of redox chemistry, trait-based functional type models of

phytoplankton have used cell size as an organizing principle – a “master trait” – for understanding phytoplankton biogeography, biodiversity, and impact on biogeochemistry^{10,26–30}. These types of theoretical constraints allow for including more functional types without introducing as many degrees of freedom as would be necessary if each were empirically described. The guiding perspective is that organizing complex biological behavior by its underlying chemical and physical constraints gives more universally applicable descriptions of large-scale biogeochemistry.”

Lines 233 – 237: This is not a particularly strenuous test of whether or not a redox-balanced approach is useful. The importance of competition with phototrophs for ammonium in limiting nitrification in surface oceans has been shown by, for instance Smith et al. (2014, PLOS One) and Xu et al. (2019, L&O) and is very easily to simulate mechanistically as long as ammonium is produced through remineralization of organic matter in both the euphotic and disphotic zones and phototrophs are given higher uptake rates for ammonium than nitrifiers.

The question is why phototrophs should have higher uptake rates for ammonium than nitrifiers in the first place. We aim to parameterize a model from first principles, that as much as possible does not use observed phenomena as input to the model. Using our methodology, it may be the yield or a combination of both yield and uptake rate that differentiate photoautotrophs from nitrifiers. Our perspective suggests that the redox chemistry explains this differentiation.

We have added a paragraph in the overview to better communicate this perspective:

L 91-101: “In contrast with empirically informed models, this approach involves constructing a model of microbial activity theoretically, and then comparing the results with the observations in order to gain an understanding of the system. The goal is to understand why biology functions as it does, in addition to anticipating global impacts. From a first-principles biogeochemical perspective with respect to physical and chemical forcing, genes are an intermediate step between forcing and function, with the detailed complexity of biological reality following the underlying chemical and physical constraints (analogous to the “form follows function” principle of architect Louis Sullivan). This does not equate to thinking that biology (or genetic information) does not matter or can be replaced entirely by physics and chemistry. Rather, we want to fundamentally understand biological activity as an integrated part of an ecosystem, and physics and chemistry become tools for doing so.”

Lines 249 – 252: I completely agree that the use of a prescribed “Martin Curve” is an inaccurate way to quantify carbon flux through the mesopelagic. My understanding is that it is seldom used except when computational limitations preclude explicit modeling of the mesopelagic ocean (e.g., in mixed layer or euphotic zone only models).

Most biogeochemical models do treat particulate fluxes and remineralization with simplified, empirically tuned parameterizations (either Martin curve or variations) because they do not explicitly model the mesopelagic. Mechanistic models of the mesopelagic that capture the complex organismal and physical processes that control remineralization present an extremely complex challenge; perhaps one of the grand challenges in modeling marine biogeochemical cycles. We have reworded this sentence as follows:

L 259-261: “In many global ocean models, the biological pump has been modeled following empirical relationships with a power law (the “Martin curve”⁸⁸).”

Lines 327-328: The question of whether genes are “intermediaries” or actively modeled is largely dependent on whether the investigator has a biogeochemical or ecological standpoint. From the biogeochemical standpoint, the gene is, in fact, only an intermediary that reflects the metabolic potential of the organism. However, from an ecological standpoint competition and selection are acting at the level of the organism (or even at the level of the gene given the prevalence of horizontal gene transfer). When considered from this perspective the gene is not an intermediary, but instead the gene (and the organism that contains it) are quite active.

It is an excellent point that genes are intermediary only from a biogeochemical perspective. We have preceded “perspective” with the word “biogeochemical” in this sentence. We have also reworked this paragraph and moved it to the overview to also emphasize that we don’t think that the genetic information is to be ignored:

L 91-101: “In contrast with empirically informed models, this approach involves constructing a model of microbial activity theoretically, and then comparing the results with the observations in order to gain an understanding of the system. The goal is to understand why biology functions as it does, in addition to anticipating global impacts. From a first-principles biogeochemical perspective with respect to physical and chemical forcing, genes are an intermediate step between forcing and function, with the detailed complexity of biological reality following the underlying chemical and physical constraints (analogous to the “form follows function” principle of architect Louis Sullivan). This does not equate to thinking that biology (or genetic information) does not matter or can be replaced entirely by physics and chemistry. Rather, we want to fundamentally understand biological activity as an integrated part of an ecosystem, and physics and chemistry become tools for doing so.”

REVIEWERS' COMMENTS:

Reviewer #1 (Remarks to the Author):

I have reviewed the response to reviewers and the authors changes to the manuscript in the revision. I am satisfied that my concerns were addressed substantively, and I feel the manuscript makes a clear case for the utility of considering redox chemistry as an underlying model principle while recognizing that it is an incomplete description of the system. I also feel satisfied with the response to the other reviewer who also raised some important points.

I strongly support publication of the revised manuscript, as I think it will benefit a broader audience than the more specialized papers on specific applications by the authors.

Reviewer #2 (Remarks to the Author):

The authors have revised their manuscript and responded to some of my concerns. However, I still do not find the overall synthesis to be particularly novel.

The authors argue for biogeochemical models that explicitly model microbes responsible for more biochemical transformations. This is in contrast to the most common current generation of models, most of which model photosynthesis, nitrate uptake, ammonium uptake, and grazing explicitly but include microbial processes associated with detritus remineralization and nitrification only implicitly through decay constants. Most biogeochemical modelers and microbial ecologists agree that explicitly modeling these processes (if done well) should make models more capable of predicting future biogeochemical responses to climate change. At the broad level, this call for more detailed microbial models is neither controversial nor novel.

The authors further argue that further progress should be organized around "redox-informed" models. Redox state and free energy have been recommended as organizing principles before. This is not a novel idea. The reason that it has not become prevalent (in the pelagic) yet is that the devil is in the details. As the authors note the free energy of a substrate determines only the biomass yield of that molecule. Determining growth rate (the key model term) requires also estimating the substrate uptake rate. This is not as trivial as the authors suggest. When limiting the discussion to nitrate and ammonium, it may be possible to use theory to predict uptake rates as the authors suggest (although even this substantially overlooks variability associated with complex cellular regulation of nitrate and ammonium transporters with varying affinity for substrates). However, the authors are arguing for broad use of this approach to model processes such as sulfate reducing bacteria, labile vs. refractory DOM, diazotrophy, and remineralization of sinking particles. For such processes substrate uptake will not be tied to substrate abundance using the same relationships as for ammonium or nitrate. For instance, sulfate reducing bacteria in the water column are likely to be found mostly within large aggregates that have anoxic interiors. Modeling these organisms in the same way that ammonium uptake by phytoplankton or nitrifying bacteria is modeled would lead to either extinction of these sulfate-reducers in the model or vastly over-estimated concentrations of sulfate. The authors argue that the average redox state of carbon atoms could also be used to structure DOM on a continuum from labile to refractory. However, this neglects complications associated with the actual uptake of organic molecules, most of which need to be broken down using extracellular enzymes before they can be taken up by bacteria. Uptake of DOM therefore cannot be simply modeled in the same way that uptake of nitrate or ammonium can. Similarly, and to an even greater extent, the authors' suggested approach will require substantial modifications when dealing with sinking particle remineralization. Particles need to find and colonize these particles, then produce extracellular enzymes to cleave specific molecules from the sinking particles in order to finally take up organic molecules that are enriched in the vicinity of the particle. Arguing that theoretical equations used to quantify nutrient uptake can in any way be extended to particles is unrealistic. Instead, it will almost

certainly be necessary to develop ad hoc parameterizations for substrate uptake of all of these different types of substrate. Since the “redox-informed” model will explicitly model growth rates as biomass yield multiplied by the ad hoc uptake rate (that will likely have to be parameterized empirically), the biomass yield will simply be subsumed into an empirical parameterization.

The authors also have a tendency to use a “straw-man” approach for arguing the benefits of their redox-informed approach. For instance, they note that this approach will improve particle flux attenuation estimates relative to “Martin-curve” parameterizations. However, “Martin-curve” parameterizations are generally only used in models that are computationally-limited and hence cannot resolve mesopelagic biogeochemistry. Any system that is computationally-limited will not be able to handle redox-informed model complexity throughout the mesopelagic. Rather, the authors should be comparing to the current version of biogeochemical models that typically explicitly model sinking particles and particle remineralization throughout the mesopelagic using the same equations employed in the euphotic zone. While these models can certainly be improved on, they are the natural comparison point, rather than models that admit that they only use the “Martin-curve” approach because they are computationally limited.

I am not against the idea of redox-informed models. I think they have potential utility and very much support research into them. However, I fail to see how this review advances the field.

Draft Only

REVIEWERS' COMMENTS:

Reviewer #1 (Remarks to the Author):

I have reviewed the response to reviewers and the authors changes to the manuscript in the revision. I am satisfied that my concerns were addressed substantively, and I feel the manuscript makes a clear case for the utility of considering redox chemistry as an underlying model principle while recognizing that it is an incomplete description of the system. I also feel satisfied with the response to the other reviewer who also raised some important points.

I strongly support publication of the revised manuscript, as I think it will benefit a broader audience than the more specialized papers on specific applications by the authors.

We thank you for your constructive comments on the manuscript.

Reviewer #2 (Remarks to the Author):

The authors have revised their manuscript and responded to some of my concerns. However, I still do not find the overall synthesis to be particularly novel.

The authors argue for biogeochemical models that explicitly model microbes responsible for more biochemical transformations. This is in contrast to the most common current generation of models, most of which model photosynthesis, nitrate uptake, ammonium uptake, and grazing explicitly but include microbial processes associated with detritus remineralization and nitrification only implicitly through decay constants. Most biogeochemical modelers and microbial ecologists agree that explicitly modeling these processes (if done well) should make models more capable of predicting future biogeochemical responses to climate change. At the broad level, this call for more detailed microbial models is neither controversial nor novel.

We are not simply arguing that microbial processes should be explicitly modeled. We agree with you that this itself is not a unique perspective. The point of our article is to propose and explain a specific guiding principle for doing so, given the complexity of the processes.

The authors further argue that further progress should be organized around “redox-informed” models. Redox state and free energy have been recommended as organizing principles before. This is not a novel idea. The reason that it has not become prevalent (in the pelagic) yet is that the devil is in the details.

We have provided and explained multiple examples in which the redox chemistry combined with population ecology has been recently demonstrated to be a useful tool for studying the pelagic. For one, it has been successfully used to articulate the relationships among nutrient concentrations, population abundances, and nitrification rates for the nitrification ecosystem in the open ocean (Zakem et al. *Nature Communications* 2018). For another, it has revealed the plausibility of stable coexistence of competing aerobic and anaerobic metabolisms in anoxic zones (Zakem et al. *ISME* 2019).

As the authors note the free energy of a substrate determines only the biomass yield of that molecule. Determining growth rate (the key model term) requires also estimating the substrate uptake rate. This is not as trivial as the authors suggest. When limiting the discussion to nitrate and ammonium, it may be possible to use theory to predict uptake rates as the authors suggest (although even this substantially overlooks variability associated with complex cellular regulation of nitrate and ammonium transporters with varying affinity for substrates). However, the authors are arguing for broad use of this approach to model processes such as sulfate reducing bacteria, labile vs. refractory DOM, diazotrophy, and remineralization of sinking particles. For such processes substrate uptake will not be tied to substrate abundance using the same relationships as for ammonium or nitrate. For instance, sulfate reducing bacteria in the water column are likely to be found mostly within large aggregates that have anoxic interiors. Modeling these organisms in the same way that ammonium uptake by phytoplankton or nitrifying bacteria is modeled would lead to either extinction of these sulfate-reducers in the model or vastly over-estimated concentrations of sulfate. The authors argue that that the average redox state of carbon atoms could also be used to structure DOM on a continuum from labile to refractory. However, this neglects complications associated with the actual uptake of organic molecules, most of which need to be broken down using extracellular enzymes before they can be taken up by bacteria. Uptake of DOM therefore cannot be simply modeled in the same way that uptake of nitrate or ammonium can. Similarly, and to an even greater extent, the authors' suggested approach will require substantial modifications when dealing with sinking particle remineralization. Particles need to find and colonize these particles, then produce extracellular enzymes to cleave specific molecules from the sinking particles in order to finally take up organic molecules that are enriched in the vicinity of the particle. Arguing that theoretical equations used to quantify nutrient uptake can in any way be extended to particles is unrealistic. Instead, it will almost certainly be necessary to develop ad hoc parameterizations for substrate uptake of all of these different types of substrate. Since the "redox-informed" model will explicitly model growth rates as biomass yield multiplied by the ad hoc uptake rate (that will likely have to be parameterized empirically), the biomass yield will simply be subsumed into an empirical parameterization.

We are not advocating specific parameterizations for uptake. We did not intend to imply that the uptake kinetics for organic matter and other substrates should be modeled identically to that of ammonia and nitrate for phytoplankton. Rather, we meant to use this as an example of how underlying physical constraints to substrate acquisition can be

exploited to develop more universally applicable descriptions for a variety of substrates and contexts. Thus this strategy can be extended to organic matter, including particles, for which other processes must be taken into account. The details of these descriptions will be different, but the guiding principle the same.

Also, we note that uptake parameterizations are not the main subject of our perspective, which is using the redox chemistry to partition consumption into biomass synthesis vs. respiration in a consistent, theoretically grounded way. This strategy can be coupled with empirical descriptions of uptake as well as theoretical ones. The physics-based descriptions are simply aligned with the overall approach of building independent descriptions of metabolism that do not rely on species- or location-specific details.

Furthermore, we note that we here are not advocating a one-size-fits-all, 'out of the box' model. Rather, given the complexity of these microbial processes, we are suggesting a principled step forwards, in contrast to succumbing to the complexity and resigning ourselves to the assumption that all must be empirically described. In this sense, we disagree that future descriptions of particle remineralization must rely on ad hoc parameterizations. We think it is a worthwhile question to inspire future research.

We have clarified our position in the manuscript:

L. 218-224: “To estimate the growth rate of each functional type, the yields from the metabolic budgets are combined with the uptake rates of the required substrates (Box 1). Limiting uptake rates may rely on empirically derived uptake kinetic parameters, or they can be estimated theoretically from diffusive supply, cell size, membrane physiology, and other physical constraints⁷⁶⁻⁷⁸. If theoretical models of uptake are used, the physical constraints on substrate acquisition and the redox chemical constraints on energy acquisition can provide an entirely theoretical estimate of the growth of each metabolic functional type.”

L. 307-312: “Much work remains in the development of these descriptions. As we discuss below, accurate estimates of organic matter turnover rates require more accurate descriptions of the complex processes governing microbial uptake rates of organic matter. However, the redox-informed yields are still useful for quantifying the relative amount of CO₂ excreted and the absolute amount of biomass sustained on a given substrate, independent of uptake kinetics (Supplementary Note 2).”

We have extended the following paragraph in the “Limitations” section and moved it adjacent to the other paragraph discussing organic matter descriptions (L. 462-472, which has also been lightly edited).

L. 448-461: “The proposed modeling approach relies on estimates of the limiting uptake rates of required substrates. In lieu of suitable theoretical descriptions, the use of empirically derived uptake kinetic parameters still employs the benefits of

the redox-informed yields (Supplementary Note 2). However, underlying physical constraints to substrate acquisition can in principle be exploited to develop more universally applicable descriptions for a variety of substrates and contexts. Uptake kinetics are complex, but for many limiting resources, encounter effectively controls the uptake, and the physics of encounter has been relatively well described. For example, uptake rates estimated from diffusive supply of substrate, cellular geometry, and membrane physiology⁷⁶⁻⁷⁸ have been empirically supported¹⁰⁷. For organic matter, future work is needed to develop suitable descriptions of consumption rates, whether empirical or theoretical. For example, descriptions require attention to the hydrolysis of organic compounds by extracellular enzymes and the ecology of sinking marine particles – the diffusion of monomer away from the particle, within-particle transport, and dynamic ecological interactions on particle surfaces, among other processes^{12,21,108,109}.”

A new supplementary note articulates the use of the yields despite uncertainty in uptake kinetics:

SI: “Supplementary Note 2:

The redox-informed biomass yield is useful even if models of substrate uptake are empirical. For a heterotroph, for example, the yield determines the amount of substrate that is transformed and excreted as a respiration product such as CO₂. In our example of the nitrification ecosystem, the redox-informed differences in the yield between ammonia and nitrite oxidizers have provided useful explanations for the observed differences between ammonia and nitrite concentrations and the biomasses of the two clades, despite uncertainty in uptake kinetics.

We can demonstrate the utility of the yield alone using a simple system for substrate concentration S and consuming biomass B :

$$\frac{dS}{dt} = S_{in} - V(S)B \quad (S2)$$

$$\frac{dB}{dt} = yV(S)B - LB \quad (S3)$$

where S_{in} is the substrate supply rate, $V(S)$ is the specific uptake rate function that depends on the substrate concentration, y is the yield, and L is the loss rate. At steady state, we can solve for the steady concentration of biomass B^* as:

$$yS_{in} = LB^* \quad (S4)$$

$$B^* = y^{-1}LS_{in} \quad (S5)$$

The steady state biomass is proportional to the inverse of the yield, and the uptake rate falls out of the equation. Therefore, the model can be useful in relating quantities of biomass to substrate supply, independent of uptake kinetics. Differences in yield alone also result in differences in the subsistence concentrations (Eqn. S1).”

We have also replaced “empirical” with “implicit” in the subheading:

L. 291: ***“Replacing implicit descriptions of organic matter remineralization”***

The authors also have a tendency to use a “straw-man” approach for arguing the benefits of their redox-informed approach. For instance, they note that this approach will improve particle flux attenuation estimates relative to “Martin-curve” parameterizations. However, “Martin-curve” parameterizations are generally only used in models that are computationally-limited and hence cannot resolve mesopelagic biogeochemistry. Any system that is computationally-limited will not be able to handle redox-informed model complexity throughout the mesopelagic. Rather, the authors should be comparing to the current version of biogeochemical models that typically explicitly model sinking particles and particle remineralization throughout the mesopelagic using the same equations employed in the euphotic zone. While these models can certainly be improved on, they are the natural comparison point, rather than models that admit that they only use the “Martin-curve” approach because they are computationally limited.

Our rationale for comparing our approach to the simplest parameterizations comes from the perspective of others that we’ve encountered that argues that there’s no point in resolving the biomass of the microbial remineralization community given its immense complexity. We assert that there is room forward on this front despite the complexity. We also wanted to address the problem to a more general audience, and thought the phrase “Martin-curve” might connect with a broader readership who might not be well versed in microbial modeling but may have come across the concept. However, we agree that computational limitations are a different problem from the limitations to the models of microbial activity themselves, so have removed this sentence.

We have revised this section as:

L. 292-312:

“Replacing implicit descriptions of organic matter remineralization

The fate of organic matter dictates the amount of carbon sequestered in the marine and terrestrial biospheres. Microbial consumption mediates the carbon stored in soils, the carbon stored in the ocean as dissolved organic matter (DOM), and the sinking flux of organic carbon that constitutes the marine biological carbon pump⁴, without which atmospheric CO₂ would be 100-200 ppm higher than current levels. We want to understand how these carbon reservoirs respond to changes in climate, such as increased temperatures and changes in precipitation patterns. However, in biogeochemical models, simple rate constants often dictate the remineralization of elements from organic back into inorganic constituents.

Replacing simplistic parameterizations with dynamic metabolic functional types means that electron-balanced descriptions of growth and respiration instead

drive the fate of organic matter in earth system models (Fig. 3). In addition to a more sophisticated and responsive description of carbon sequestration, non-living organic matter is fully integrated into ecosystem frameworks, enabling theoretical studies of phytoplankton-bacteria interactions to complement observational and experimental approaches.

Much work remains in the development of these descriptions. As we discuss below, accurate estimates of organic matter turnover rates require more accurate descriptions of the complex processes governing microbial uptake rates of organic matter. However, the redox-informed yields are still useful for quantifying the relative amount of CO₂ excreted and the absolute amount of biomass sustained on a given substrate, independent of uptake kinetics (Supplementary Note 2).”

I am not against the idea of redox-informed models. I think they have potential utility and very much support research into them. However, I fail to see how this review advances the field.

Draft Only